# Treatment of Pediatric Anogenital Warts in the Era of HPV-Vaccine: A Literature Review

**DOI:** 10.3390/jcm12134230

**Published:** 2023-06-23

**Authors:** Astrid Herzum, Giulia Ciccarese, Corrado Occella, Lodovica Gariazzo, Carlotta Pastorino, Ilaria Trave, Gianmaria Viglizzo

**Affiliations:** 1Dermatology Unit, U.O.C. Dermatologia e Centro Angiomi, IRCCS Istituto Giannina Gaslini, Via Gerolamo Gaslini 5, 16147 Genova, Italy; corradooccella@gaslini.org (C.O.); lodovicagariazzocelesia@gaslini.org (L.G.); carlottapastorino@gaslini.org (C.P.); gianmariaviglizzo@gaslini.org (G.V.); 2Dermatology Unit, Department of Medical and Surgical Sciences, University of Foggia and Ospedali Riuniti, Viale Luigi Pinto, 71122 Foggia, Italy; giulia.ciccarese@unifg.it; 3Department of Dermatology, DISSAL, University of Genova, 16132 Genova, Italy; ilaria.trave@unige.it

**Keywords:** HPV, pediatric, genital warts, podofillotoxin, HPV-vaccination

## Abstract

Anogenital warts (AWs) represent a therapeutic challenge, especially in infants, due to sensitive skin and frequent disease recurrence. Though the initial wait-and-see approach is often adopted in asymptomatic immunocompetent children, with spontaneous clearing in almost 90% of cases within two years, persistent or symptomatic lesions can be reasonably treated. However, few studies have been conducted on children. Consequently, most treatments on patients under age 12 are not approved by the Food and Drug Administration. Herein, we review possible therapies for pediatric use in AW and report an illustrative case of a two-year-old boy with atopic skin and symptomatic, persistent AWs who was successfully treated with topical podophyllotoxin, without adverse effects or recurrence. Among available therapies for AWs, topical therapies, such as immunomodulating-agents (topical imiquimod 5% and 3.75% cream, sinecatechins 15% ointment) and cytotoxic agents (podophyllotoxin and cidofovir) are considered manageable in children because of their low aggressiveness. In particular, podofillotoxin gel 5% and imiquimod 5% cream have been reported to be safe and efficacious in children. Currently, HPV vaccination is not recommended as a treatment for established HPV infection and AWs, yet a possible therapeutic role of HPV vaccination was recently suggested in the literature and deserves mention.

## 1. Introduction

Human papillomavirus (HPV) belongs to the family of Papillomaviridae, which are non-enveloped double-stranded circular DNA viruses with oncogenic potential [1].

With regard to their oncogenic potential on infected epithelia, HPVs are classified into low-risk -(LR-) and high-risk (HR-) oncogenic genotypes [1,2]. LR-HPV genotypes present lower oncogenic potential and mainly induce benign epithelial proliferations. LR-HPV genotypes include HPV types 1, 2, 3, 4, 10, 26, 27, 28, 29, and 41, which are mainly associated with the development of cutaneous warts in infected skin [3,4,5]. Additionally, HPV 6 and 11, which are linked to the development of anogenital warts (AWs), are included among LR-HPV genotypes. Indeed, up to 90% of AWs are caused by HPV genotype 6 and 11 [2]. Other LR-HPV genotypes associated with AWs are HPV 40, 42, 43, 44, 54, 61, 72, 81, and 89 [3,4,5]. On the other hand, HR-HPV genotypes include HPV 16, 18, 31, 33, 35, 39, 45, 51, 52, 55, 56, 58, and 59, which harbor high oncogenic potential, with high-risk of inducing oncogenesis with the development of high-grade dysplasia and even invasive malignant squamous cell carcinoma of infected skin and mucosal surfaces, including vulva, vagina, cervix, penis, anus, and oropharynx, in cases of persistent infections [6,7,8,9,10,11,12]. Among these, cervical squamous cell carcinoma accounts for most HPV-related cancers, and, notably, 70% of cervical carcinomas are caused by HPV, mainly types 16 and 18, with a higher prevalence of HPV genotype 16 [2]. Remarkably, cervical carcinomas represent the fourth most common cancer in women worldwide [2,9].

The genome of HPV consists of six early genes (E1, E2, E4, E5, E6, E7) that encode proteins that are synthesized during initial infection and two late genes (L1 and L2) that encode proteins that are produced later on, after the synthesis of early proteins [7,12,13]. Early viral proteins encoded by E genes stimulate cell proliferation. Additionally, oncoproteins E6–E7 are included among early viral proteins. These promote the replication of viral DNA and prevent host cell apoptosis. Among late structural proteins, L1 represents the major viral capsid protein. It influences tumorigenicity and epitheliotropism. L2 brings about the formation of progeny virions, thereby accomplishing the viral life cycle [7,12,13]. After viral transmission, HPV can cause latent infection, remaining in the nucleus of host cells as a double- stranded DNA episome, thereby prolonging the incubation period after the initial contact with the virus for several months [1,7]. Conversely, during a productive infection, replication of the viral genome through the host cell DNA polymerase is enhanced by stimulation of keratinocyte proliferation, which is, in turn, induced by viral E proteins. [1,7] In detail, in active warts, the virally induced increase in keratinocyte numbers favors hyperkeratosis with a thickening of the granular, spinous, and basal cell layers of the epidermis, allowing the virus to spread through skin layers and ultimately to be released from dead cells detaching from upper skin layers [1,5]. Specifically, the development of a wart takes, on average, three to four months of virally enhanced keratinization, induced by viral E proteins, as previously described [1,5]. The resulting AWs are highly contagious, with a transmission rate of 65% during sexual contact [2]. Not surprisingly, the incidence of AWs is rising, both in adults and children; the increasing incidence in children could be the consequence of the concomitant increase in prevalence among caregivers [2]. Undeniably, HPV infection stands out as the most common sexually transmitted infection (STI), affecting the majority of sexually active individuals at least once in their lifetime [1,7] and negatively impacting on the quality of life, mood, and sexual function [8]. AWs are a relevant pediatric issue, though recent data showed a reduced incidence of AWs in pediatric patients and young adults after the introduction of the HPV vaccination [4].

Notably, young children may acquire HPV during childbirth from the mother’s genital tract, or during diaper care and when bathing, as a result of active outbreaks of warts on the caregiver’s hands or poor hand hygiene; however, an evaluation of all possible means of transmission, including sexual abuse, is required.

A detailed multidisciplinary evaluation is required when managing AWs, including a complete physical and psychological examination, with particular attention to genital areas, always considering the possibility of sexual abuse [1,4,7]. Among different forms of sexual abuse, genital-genital contact, oro-genital contact, genital-anal contact, digital penetration of the anal canal and/or of the vaginal tract, the introduction of objects into the vagina or anus, and caressing are investigated in most studies handling this topic [1].

Especially in young children, treatment is complicated by sensitive skin and frequent recurrences [1,2,3,4,5,6,7,8,9,10,11,12,13,14]. Particularly in infants with atopic dermatitis, AWs represent a therapeutic challenge, as the weak skin barrier facilitates cutaneous infections and viral persistence, in addition to hindering aggressive therapeutic methods [14,15]. Of note, a high rate of spontaneous clearing is reported in immunocompetent children, namely, 70% in one year and 90% in two years, prompting the adoption of a wait-and-see approach in asymptomatic AW cases [1,2,15]. However, immunosuppression and atopic skin may favor viral persistence and thereby facilitate secondary infections, symptomatic disease, and even thriving HPV infections, including Buschke-Lowenstein tumors, giant tumor masses largely occupying the anogenital area, Bowen disease, and acquired epidermodysplasia verruciformis [14,15,16,17,18]. Accordingly, persistent lesions, lasting over 2 years, or symptomatic lesions, associated with pruritus, burning, bleeding, and secondary infections, are actively treated [19]. Treatment options include surgical interventions, often requiring anesthesia, and non-surgical therapies, such as topical preparations [19]. Considering that children have low pain tolerance, it is advisable to approach pediatric AWs with low aggressiveness, as can be achieved with repeated domiciliary mild applications of topical therapy and combination therapies, always to be discussed with the caregiver [1,19]. However, currently, no topical treatment for AWs is licensed by the Food and Drug Administration (FDA) for patients under age 12, as only a few studies have been conducted on children to date [1]. Indeed, in the literature, only a few reports of pediatric use of topical preparations can be found, reporting similar inter-category efficacy and similar low impact in terms of local skin reactions, including inflammation with edema, erythema, erosions, burning, pain, and itching [1,20].

## 2. Materials and Methods

Herein, we provide a review of the literature regarding currently available therapies for pediatric AWs and report an illustrative case of a two-year-old child with atopic dermatitis and AWs treated with topical podophyllotoxin.

We performed a literature search of English reports on topical therapies available for the treatment of AWs in children on PubMed. To retrieve all relevant articles, the following terms were researched: “warts”, “anogenital warts”, “HPV”, “human papillomavirus”, “pediatric”, “children”, “childhood”, “treatment”, “therapy”, and “topical”.

There was a special emphasis on the treatment of young (<12 years) children, keeping in view the fact that great difficulties exist in the treatment of this age group, as no topical treatment for AWs is licensed by the FDA for patients under age 12, and other, more aggressive therapies are poorly tolerated.

## 3. Results

Numerous therapeutic approaches are available for the treatment of AWs, with the primary aim of eliminating visible warts, yet with possible recurrences, as therapy does not clear the causative viral infection itself [1,2]. Treatments include patient-applied topical therapies and healthcare provider-administered topical, intralesional, and physical destruction methods [1,2]. Though treatment options are numerous, in children under 12 years of age, most treatments are not approved, and especially in young children, treatment methods should be discussed with the caregiver and possibly combined [1,2].

### 3.1. Patient-Applied Topical Therapies

Patient-applied topical therapies are reported to be well tolerated in children because of the low aggressiveness of single applications; however, these necessitate repeated sessions to achieve therapeutic results and can give rise to complications [1,2]. These therapies comprise immunomodulating agents (topical imiquimod 5% and 3.75% cream, sinecatechins 15% ointment) and cytotoxic agents (podophyllotoxin and cidofovir) [1,2].

In detail, topical podophyllotoxin 0.5% is an antimitotic agent derived from purified podophyllin resin that inhibits the formation of microtubules. It is approved for the treatment of anogenital warts in adults (topical 0.5% formulation), with a reported clearance rate of 45–88% and a recurrence rate of 12–60% [3,19,21]. Treatment consists of two daily applications for three days, followed by four days’ rest, repeating this cycle 4–5 times [3,21].

In children, minimal irritation was reported with topical podophyllotoxin 0.5% application while still obtaining therapeutic efficacy. Moresi et al. reported 88% resolution of AWs in 17 children <2 years, without recurrence within 2–16 weeks of the use of podophyllotoxin 0.5% gel at the reduced dosage (tapered single week application) [19]. Furthermore, Stefanaki et al. reported 80% resolution of AWs in 20 children (1–11 years) with topical podophyllotoxin 0.5% applied twice daily for 3 consecutive days followed by 4 days’ rest [3]. Among common side effects, burning and pain, but also erythema, edema, erosions, and bleeding were reported [3,19].

Imiquimod is an imidazoquinolin heterocyclic amine that induces the clearance of warts through local antiviral and antitumor properties, acting as a toll-like receptor-7 (TLR-7) agonist and stimulating local macrophages to produce cytokines, particularly interferon-α (IFN-α). Treatment is applied three times weekly once a day and is then left to act on the skin for 6–10 h, for up to 8–16 weeks [3,21].

Local inflammatory side effects are common, especially erythema, with itching, vesiculation, and ulceration, while flu-like symptoms occur less often [2,21]. In adults with AWs, clearance rates of 37–72% were reported with imiquimod 5% cream, with 13–19% recurrence rates [21]. Though not FDA-approved in children younger than 12, Imiquimod was reported as safe and effective in treating common warts and AWs in young children in retrospective case series and in single reports [19,22,23,24]. In detail, in a study on eight children, (two patients < 2 years), treated thrice weekly for 2–4 months, clearance was reported in 75% with no recurrences at 6–12 months. Among side effects, irritation and pruritus were predominant, yet 50% experienced few or no side effects [19]. Additionally, single reports described the clearance of AWs with imiquimod 5% cream: in a 6-month-old girl within 3 weeks of treatment; in a 19-month-old girl within 5 weeks of treatment; and in a 2-year-old boy within 5 weeks of treatment [22,23,24].

Sinecatechins are a green tea extract that was approved by the FDA in 2006 as a treatment for AWs with thrice daily application for up to 16 weeks in adult patients (>18 years) [25]. In adults, the application of 10% sinecatechins ointment obtained clearance rates ranging from 45.5 to 64.9%, with occasional mild local skin reactions and 6–11% recurrence rates [21,25,26]. Only limited data exist for children, including the successful treatment of an 11-year-old boy with AWs with 10% sinecatechins ointment [27]. Godoy-Gijon et al. described a 7-year-old girl with a history of atopic dermatitis, previously unsuccessfully treated with six cycles of cryotherapy and topical 5% imiquimod cream for 16 weeks. The girl experienced complete clearance of AWs after three times daily application of a sinecatechin ointment 10% for 9 weeks. No adverse reactions or recurrences in one-year follow-up were reported [28].

Cidofovir is an antiviral agent used to prevent the replication of DNA viruses by blocking DNA polymerase. AW therapy with topical cidofovir 1–3% cream is rarely reported in children: one report described the clearance of resistant AWs, which had been non-responsive to cryotherapy, imiquimod, and podophyllotoxin, in a 6-year-old-girl treated cidofovir 3% cream [29]. Additionally, a 3-year-old boy was successfully treated with cidofovir 1% cream daily for 5 days every week for two weeks and then after one month. No recurrence was reported after one year [30]. However, it must be considered that cidofovir is potentially nephrotoxic, especially in patients with renal failure, and is expensive and scarcely available, limiting its use in general practice in the treatment of resistant AWs [30].

### 3.2. Healthcare Provider Administered Topical Therapies

Other topical therapies include healthcare provider-administered cytotoxic agents such as trichloroacetic acid (TCA), podophyllin resin, and nitric-zinc complex (NZC). [2,21].

TCA 80–90% induces cell death by protein coagulation. It is applied weekly by health-care providers, directly onto warts, avoiding spreading the solution onto the surrounding skin, where it can cause ulceration. In adults, clearance rates of 70–81% were reported, with a recurrence rate of 36% [21]. The use of TCA has been scarcely reported in children with anogenital warts: Varma et al. reported the case of a 9-year-old girl with multiple AWs treated with TCA at 80% on three separate occasions 2 weeks apart, with marked improvement at each visit [31].

*Podophyllin* resin, administered by healthcare providers at 10–25%, is derived from the root of *Podophyllum* sp. and carries a risk of systemic side effects, including neurotoxicity, liver dysfunction, and bone marrow suppression. In adults, 38–79% response rate and 21–70% recurrence rates were reported. In children, the use of podophyllin resin is not recommended [21].

Furthermore, 25% podophyllin in a benzoin tincture is available for the removal of soft genital AWs, though this is not FDA-approved. It is administered by healthcare providers. To avoid systemic absorption, dried 25% podophyllin should be kept in place for 1 to 4 h, then thoroughly removed.

Provider-applied topical NZC also deserves to be mentioned. It is a recently developed topical solution containing 65% nitric acid, zinc, copper, and organic acids, exerting a caustic effect on AWs, with an 84–87% cure rate in wide population studies in adults [32,33]. Ciccarese et al. observed relapses in 29% of patients at 3 months and in 5% at 6 months, with better responses in patients with ≤ 5 warts [32]. In children, it was administered only for cutaneous warts by Giacaman et al. [34].

Finally, Interferon represents a healthcare provider-administered therapy; it can be used topically and intralesionally for treatment-resistant genital warts. However, it is not recommended as a first-line treatment because it is expensive, has uncertain effectiveness, and has not been tested on children [2].

### 3.3. Healthcare Provider-Administered Physical Destruction Methods

Among other healthcare provider-administered methods, physical destruction methods include cryotherapy, surgical removal, and laser vaporization [2]. These are poorly tolerated by children because of treatment-related pain, making general anesthesia usually necessary in younger children [2]. Furthermore, postoperative pain and scarring can occur. For these reasons, physical destruction methods are mainly reserved for children with extensive AWs [2]. Clearance rates range from 27% to 100% in adults, with a recurrence rate of approximately 25% [35]. In detail, cryotherapy is an inexpensive treatment that is highly user dependent in which liquid nitrogen is applied directly to warts. It is most effective on keratinized warts, solitary or in small numbers, and clearance is mostly obtained after several treatments. In adults, clearance rates of 54–88% have been reported, with 21–40% recurrence rates [21]. However, in younger children, it is not well tolerated because of pain. Other common side effects include local skin irritation and blistering, followed by hypopigmented sequelae [2,21]. Surgical techniques include shave and radical excisions, with a remission rate of 36–100% in adults and 8–65% recurrence [21], and electrosurgery, which applies electrical current directly to warts, causing pain and scarring as common side effects; this technique obtains 64–94% clearance and up to 50% recurrence [21]. Surgery is indicated for large, sessile, keratinized warts with a base wider than 1 cm, or extensive treatment-resistant warts. Additionally, when a histological diagnosis is needed, excision is considered the preferred treatment option [2,21]. Pulsed-dye lasers and carbon dioxide (CO_2_) lasers are also used to treat resistant anogenital warts but are associated with pain and scarring and their availability is scarce. In adults, CO_2_-lasers yield 40–100% response rates and 4–77% recurrence rates [21]. Therefore, laser treatments are not considered first-line therapy [2,21,36].

### 3.4. Healthcare Provider-Administered Intralesional Therapies

Intralesional immunotherapies, including *Candida* antigen, tuberculin/purified protein derivative (PPD), bacillus Calmette-Guerin (BCG) vaccine, and the measles, mumps, and rubella (MMR) vaccine, have been proven safe and effective for the treatment of cutaneous warts in children [37].

Additionally, for the treatment of AWs, these intralesional immunotherapies are promising and cause only mild and transient adverse effects, including localized inflammation at the injection site and flu-like symptoms due to the induced [38] release of Th1 cytokines (IL-2, IL-12, TNF-α, IFN-γ) that activate CD8 cytotoxic T lymphocytes and NK lymphocytes to eradicate HPV infection [39,40].

Recently, Nofal et al. reported complete regression of multiple AWs in 80% of children after *Candida* antigen injection and 73% after MMR injection [41].

### 3.5. Healthcare Provider Administered HPV-Vaccination

Currently, HPV vaccination is not recommended as a treatment for established HPV infection and HPV-related disease, yet a possible therapeutic role of HPV vaccination was recently suggested in the literature and would plausibly, if confirmed, represent the beginning of a new era for the management of HPV-induced AWs [42]. Indeed, a possible therapeutic role of HPV vaccination could be of great relevance, especially in pediatric patients that poorly tolerate the pain related to other, previously described AW treatments [21], and in difficult-to-treat patients, such as patients with atopic dermatitis and sensitive skin, and immunosuppressed patients, who are prone to developing persistent HPV infections and generalized AWs [14,15].

Indeed, a relevant effect on AWs of worldwide-adopted HPV vaccination programs has been observed already [4]. Recent data indicate that after the introduction of the HPV vaccination, the incidence of AWs has already dropped dramatically, both in children and adults, reaching a 72% decrease among 16–18-year-old girls and a 51% decrease among boys [4].

HPV vaccines were originally developed as a primary prevention measure to reduce HPV infection rates and HPV-related tumors and were rapidly established as a successful prevention measure of anogenital cancer and AWs. In 2006, the FDA approved a quadrivalent HPV vaccine (Gardasil) that targets HPV genotypes 6, 11, 16, and 18. A bivalent HPV vaccine (Cervarix) targeting only HR-HPV genotypes 16 and 18 was approved in 2009 [42]. In 2014, a nonavalent HPV vaccine (Gardasil-9) targeting HPV genotypes 6, 11, 16, 18, 31, 33, 45, 52, and 58 was licensed. Bivalent, quadrivalent, and nonavalent HPV vaccines are all administered intramuscularly and are based on the type-specific HPV L1 major capsid protein, assembled to form recombinant, non-infectious virus-like particles (VLPs) which are antigenically identical to infective virions [42]. VLPs elicit the host’s subtype-specific neutralizing antibody response, which inhibits viral entrance into epithelial cells and possibly confers also a degree of cross-protection against non-vaccine-specific subtypes [42]. This cross-protection would occur in addition to the well-known type-specific humoral response elicited by the HPV vaccine, which activates B memory cells that produce type-specific antibodies [43]. Furthermore, HPV vaccination was linked to increased levels of TNF-α, IL-2, and proinflammatory cytokines (IL-1α, IL-1β, and IL-6) and to the induction of IFN-γ-producing cytotoxic cells which are needed to clear viral infection. These local alterations of the cytokine-microenvironment might play an additional role in the clearance of already established HPV-persistent infections and HPV-induced proliferation [43,44].

It must be specified that, so far, controversial results about a possible therapeutic role of HPV vaccination have been reported. Kreuter reported no therapeutic efficacy of quadrivalent HPV vaccination in adults and pediatric patients with AWs [45,46].

Conversely, numerous literature data suggest a possible therapeutic effect of HPV vaccination on existing HPV infection and HPV-induced benign proliferations, including AWs and cutaneous warts (CW) [47,48,49,50,51,52]. Regarding AWs, only Couselo Rodriguez treated a child with Gardasil-9, obtaining partial remission [53]. In adult AWs, the administration of the HPV vaccine achieved remission in 3 out of 3 patients, as reported by Dianzani, 6 of 10 patients, as reported by Choi, 1 of 1, as reported by Lee, 2 of 5, as reported by Bossart, and 1 of 1, as reported by Moscato [54,55,56,57,58].

The exact mechanism influencing the therapeutic effect of HPV vaccines has not yet been explored. The efficacy of commercially available HPV vaccines is based on their ability to induce a humoral immune response [43,59]. However, the regression of cervical, anal, and vulvar lesions has been associated with activation of cellular, rather than humoral, immunity [60].Yet, while the preventive role of HPV vaccination in HPV-naive subjects has been established, the role of HPV vaccination on subjects already infected with HPV remains unclear, as well as its therapeutic effect on persistent HPV infection and on clinically evident viral-induced epithelial proliferations [61,62,63,64,65,66].

Systematic studies analyzing the possible therapeutic effects of HPV vaccination on AWs and other HPV-induced epithelial proliferations are required to gain a deeper understanding of the still puzzling immune mechanisms underlying the possible clearance of HPV-related lesions after HPV vaccination and to establish a potential therapeutic role of the vaccine in recalcitrant HPV-proliferations [62]. Indeed, the prospect of possibly achieving both primary prevention as well as targeted HPV treatment by administering HPV vaccines encourages us to invest in research in this field.

### 3.6. Case in Point Report

A two-year-old boy presented at the Dermatology Unit with proliferative, warty, slightly pruritic, perianal tumors which had been worsening for one year. The child, delivered vaginally at term, had a history of atopic dermatitis, hyper-eosinophilia, and food allergy to eggs, fish, and bananas.

Clinical examination evidenced multiple perianal, hyperkeratotic, skin-colored papules 1–3 mm in diameter, partially coalescing into plaques; no mucosal alterations were evidenced (Figure 1). Diagnosis of AWs was established clinically and complete skin and mucosal examination excluded erythema, bruises, lacerations, and erosions. At anoscopy, the anal canal was spared, without anal sphincter abnormalities or intra-anal mucosal alterations. Multidisciplinary evaluation excluded signs of physical injury or abuse, and screening for STIs including HIV, hepatitis, and syphilis tested negative. Of note, cutaneous warts were evidenced on the caregiver’s finger (Figure 1).

The child was administered topical podophyllotoxin 0.5% once a week for 30 days by the caregiver at home, obtaining complete clinical remission of AWs after one month of therapy, without irritation or other adverse effects. Additionally, at clinical follow-up after 3 and 6 months, no recurrences were evidenced.

Written informed consent to the publication of case details was obtained.

## 4. Discussion

HPV is an epitheliotropic virus, specifically infecting cutaneous surfaces and mucosae, including the vulva, vagina, cervix, penis, anus, and oropharynx [67,68,69,70]. Clinically, infection can be asymptomatic or can cause epithelial benign proliferations, such as cutaneous warts and AWs (mainly associated with LR-HPV genotypes 6 and 11) and malignant proliferations, including cervical cancer (mainly associated with HR-HPV genotypes 16, 18, 31, 33, 35, 39, 45, 52, 55, 56, and 58) [67,68,69,70,71,72,73].

Accordingly, AWs represent benign epithelial proliferations associated mainly (90%) with LR-HPV infection genotypes 6 or 11, while in up to 10% of cases, HR-HPV coexists, possibly leading to the development of malignant squamous cell carcinoma in a minority of cases [71]. Moreover, AWs can coexist with other HPV-related diseases, including intraepithelial neoplasia of the perianal skin (PAIN), anus (AIN), penis (PIN), vulva (VIN), and vagina (VAIN), as well as possibly coexisting with malignant squamous cell carcinoma [71]. Typically, after exposure to HPV, the incubation time of HPV infection ranges between 3 weeks and 8 months, with most AWs appearing within 2–3 months after exposure [2]. With an annual incidence of 0.15% in adults and a prevalence of 0.2% to 5.1% at genital examinations, AWs rank first among sexually transmitted diseases, representing 38.7% of all diagnosed STIs (40.7% in males, 34.2% in females) [70,71,74,75]. The highest rate of AWs is recorded in 16–24-year-old females and in 20–24.3-year-old males, pointing out a peak in incidence, especially in young, sexually active individuals, and earlier in females than in males [70,71,74]. Notably, Western countries have experienced an epidemiologic growth trend of AWs in recent years, possibly associated with growing promiscuity, a higher number of lifetime sexual partners, and early age at first intercourse [71,74]. Fortunately, the epidemiological picture lately seems to reflect the influence of HPV vaccination, and especially in young people, incidence decreases of AWs have been reported [71]. Clinically, AWs mostly present as multiple, clustered, flesh-colored or hyperpigmented, small (<5 mm) papular lesions, possibly coalescing into plaques, while the occurrence of a single AW is rarer [71]. Lesions are mainly asymptomatic, yet slightly symptomatic lesions can be observed, especially as a consequence of irritation and bleeding or secondary infections, with pruritus or burning sensation representing the most commonly reported symptoms [71].

Regarding the prevalence and persistence of the causative HPV infection, data are disparate and highly influenced by anatomic site, age, sex, HPV genotype, and other risk factors for viral persistence, including number of sexual partners and smoking [69]. In a recent study conducted on women attending the Dermatology Unit of a reference hospital in Italy, cervical, anal, and oral HPV infection was detected respectively in 48%, 26%, and 29% of women screened [68]. Furthermore, recent studies by the Center for Disease Control and Prevention (CDC) reported a 42.7% prevalence of cervical HPV infection and 45.2% prevalence of penile HPV infection in the USA [72,73]. Prevalence of HPV infection increases in women with abnormal cervical pathology in proportion to the severity of cervical lesions, reaching up to 90% in women with grade 3 cervical intraepithelial neoplasia and invasive cancer [70].

Remarkably, anogenital HPV infection is mainly transmitted sexually in adults, occurring in more than 40% of sexually active women and representing the most common STI [1,67,71]. Conversely, the association of HPV infection with sexual activity varies greatly with age, and in young children, association is less frequent than previously thought [76]. In any case, approaches to treating pediatric AWs need to be chosen with care and address both the child and the family, starting with a fundamental detailed history of the child and the caregivers, giving specific relevance to genital and non-genital warts presented by caregivers and to possible genital HPV infection of the mother [1]. Additionally, a general physical examination aimed at assessing possible signs of abuse is mandatory in children, with particular attention to the genitals, anus, and perineum. Even the anal sphincter must be examined with anoscopy to assess possible anal sphincter abnormalities and intra-anal mucosal alterations. Other findings indicative of possible abuse include physical injuries such as bruising, ecchymoses, petechiae, acute trauma of the anogenital tract, with anal and vaginal lacerations and tears, vaginal/penile or anal bleeding or pain, vaginal/penile or anal discharge, recurrent urinary infections, hymen rupture or enlargement of its transverse diameter, or even pregnancy in young girls [1,20]. Notably, according to the long incubation time of HPV infection, abuse might have taken place months prior to the time of the visit, and acute signs of abuse might not be visible anymore [2,20]. Indeed, in children with STIs, clinical findings are mostly unremarkable or non-specific during anogenital examination [2]. Therefore, a global psychological evaluation and an evaluation of social risk factors are needed as well as the exclusion of other possible STIs with serological testing for HIV, syphilis, hepatitis B and C and analysis of any anogenital exudate [20,76].

Although the identification of pediatric anogenital warts should always alert physicians to the possibility of sexual abuse, it must not be considered pathognomonic. Indeed, in young children (<4 years) with no other STI and no evidence of anogenital trauma or clinical indicators of physical abuse, HPV is most likely non-sexually acquired [76]. Girls are three times more frequently affected by AWs than boys and, though the perianal distribution of AWs might seem suspicious, it is actually the most common localization of AWs in young (<2 years) children, as in the case presented above, even in the absence of sexual abuse, while the penile shaft, urethra, and vulva are usually unaffected [2,76]. The latter become preferred sites of AWs, together with perianal localization, only from later childhood and adolescence [2,76].

HPV transmission in young children is hypothesized to occur either “vertically” or, more predominantly, “horizontally” [2,76]. Vertical “periconceptive” transmission, through infected spermatozoa or oocytes, has been only theorized, as well as “prenatal” transmission in utero, possibly occurring through fetal membranes, amniotic fluid, and umbilical cord blood [1,76,77]. Principally, vertical transmission occurs “perinatally” during childbirth by direct contact with an infected maternal genital tract. This occurs especially if the cervical viral load is high, possibly resulting in diffuse AWs and laryngeal papillomatosis of the newborn [78]. Vertically transmitted HPV is mainly diagnosed within the first two years of life, though variable viral latency hinders the determination of exact upper age limit for this type of transmission [76]. In any case, HPV is mostly transmitted horizontally, i.e., by direct contact through hetero-inoculation from caregivers, even non-sexually and non-genitally, or through auto-inoculation from other infected body sites of the child itself.

Of note, children can become affected during diaper care and when bathing as a result of active outbreaks of warts on the caregiver’s hands’ or poor hand hygiene. Furthermore, indirect horizontal transmission through towels, linings, and personal hygiene products has been theorized, though this route is considered to have little impact [1].

Notably, in atopic dermatitis, impaired local immune response and altered skin barrier may favor viral infection and persistence [16,17]. Additionally, viral infection and persistence may be favored by immunosuppression, both iatrogenic and non-iatrogenic, especially in organ transplant recipients, where cell-mediated immunity is reduced, as well as in diabetic patients or after repeated local trauma [16,17,18,78].

Recently, the previously mentioned potential therapeutic effect of HPV vaccination on AWs has raised interest in this research field due to the compelling prospect of possibly obtaining both primary HPV prevention as well as targeted HPV treatment [60,61,62,63,64,65,66]. Especially in the pediatric population, this prospect seems encouraging, as HPV vaccination has been reported to be more effective in young individuals than in adults [79]. Indeed, after HPV vaccine administration, youngsters generate higher numbers of HPV-specific memory-B lymphocytes and T lymphocytes compared to adults, leading to enhanced immune response. [79]. Furthermore, after puberty, post-puberal hormones cause the expression by HPV-infected cells of major histocompatibility complex class I (MHC-I) to decline, negatively impacting the activation of HPV-specific immune-response mediated by cytotoxic-T cells [80].

Especially in children, a potential dual role of HPV vaccination, i.e., therapeutic and preventive, seems plausible and compels us to obtain in-depth insight into possibly unexplored potentialities of the HPV vaccine. Yet, to date, the authors note that HPV vaccination is not recommended as a treatment for established HPV infection and HPV-related diseases such as AWs [42]. The main role of HPV vaccination remains primary HPV prevention, yet even just through prevention programs, a relevant effect of HPV vaccination on AWs has already been observed. Literature data show a reduction of AW incidence of up to three-quarters in young girls, to whom vaccination programs are mainly addressed, and a reduction of AW incidence by one half in young boys [4]. Conceivably, this recently observed decrease in AWs can be interpreted as a result of the reduced HPV infection rates among the population due to the primary prevention provided by HPV vaccination [4].

In the meantime, HPV-therapeutic vaccines are also under trial for use in cervical cancer and cervical intraepithelial neoplasia (CIN), AIN, and VIN, but at present, no HPV-specific antiviral treatment is available, nor can therapies adopted to remove AWs reliably eradicate HPV infection [42,50]. Indeed, it is well known that after treatment of AWs, latent HPV can persist in epithelial tissues and can lead to recurrences in up to 70% of cases within 6 months in apparently successful therapy [20].

Consequently, to date, considering the high number of recurrences due to latent viral infection, only limited cases of AWs warrant a therapeutic approach, as in the described illustrative case, which presented symptomatic AWs that worsened for one year. In our experience, as highlighted by the presented case in point, at-home topical therapy is preferred as an alternative to more aggressive healthcare provider-administered therapies. Indeed, it must be considered that especially young children have difficulties in tolerating the pain related to healthcare provider-administered therapies [1,2,3]. Furthermore, the choice of administering topical podophyllotoxin 0.5% in our clinical practice is supported by recent reports of its therapeutic efficacy (80–88%) with minimal irritation in children [3,19]. Indeed, among the patient-applied therapies described for pediatric use, podophyllotoxin gel 5% and imiquimod 5% cream were notably reported to be both safe and efficacious in children [19,22,23,24]. Recently, Sinecatechins have been reported to have good clinical outcomes in children, yet the literature on this topic is still theoretical, and administration thrice daily is still considered impractical [25,26,27,28]. Topical Cidofovir was excluded because of its limited availability and high cost [2,29,30].

## 5. Conclusions

Conclusively, a patient-tailored approach is necessary when dealing with AWs, especially in young children, whose history might be complex and for whom a cure might be impaired by sensitive skin and frequent recurrences [1,2,3,81]. First of all, an attentive evaluation of the physical signs possibly associated with AWs in children is necessary, as well as an assessment of possible conditions that may complicate therapeutic compliance and local healing.

Additionally, it is important to consider that children have low pain tolerance, possibly influencing their compliance to some therapies, mainly aggressive ones. Therefore, pediatric AWs should be treated with low aggressiveness therapies, such as mild repeated domiciliary topical therapies that can be effective and safe in pediatric Aws, even if atopic dermatitis and related skin barrier defects coexist, representing a valid alternative to painful, aggressive treatments.

Lastly, though HPV vaccination is not recommended as a treatment for AWs, a possible therapeutic role of HPV vaccination has been reported which highlights the need to obtain a more in-depth understanding of the possibly unexplored therapeutic potentialities of the HPV vaccine.

## Figures and Tables

**Figure 1 jcm-12-04230-f001:**
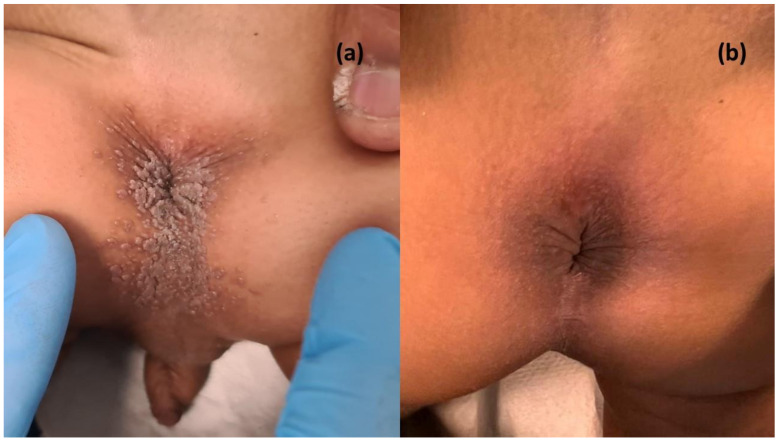
(**a**) Multiple, proliferative, warty perianal warts of a two-year-old atopic patient; note the cutaneous wart of the caregiver’s finger. (**b**) Complete clinical remission of anal warts after one month of therapy with topical podophyllotoxin 0.5% once a week.

## Data Availability

The data that support the findings of this study are available from the corresponding author, [A.H.], upon reasonable request.

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
