# Peer review of "Treatment of Pediatric Anogenital Warts in the Era of HPV-Vaccine: A Literature Review"

_jcm, 2023, doi:10.3390/jcm12134230_

Round 1

Reviewer 1 Report

Thank you for the opportunity to review this case report and review of the literature on management of anogenital warts in the pediatric population.

I have the following comments:

The authors while describing many treatments discussed in case reports in the literature, they focus on podophylin. There are several podophylin formulations, some of which are not approved by the FDA (PODOCON 25 - 25% Podophylin in benzoin tincture and intralesional pdophyllin). It may be important to highlite this.

It is not clear in the case report if the topical podophyllotoxin was applied by the parent/caregiver or the healthcare provider.

While the authors highlite delivery through a birth canal affected by condyloma and sexual abuse as situations where a child can become infected, there is another situation. If the caregiver has an active outbreak of genital wart and poor hand hygiene, then in doing diaper care ect, the child can become affected. This may be elluded to but not clearly documented in paragraph lines 77-85.

Issues related to english language:

1. Abstract - Paragraph 2 - "without adverse effects nor recurrences" means that there were recurrences because its a double negative. The correct word is "or".

2. Introduction - paragraph 5 - best not to use words like "moreover" at the start of a paragraph. 

Issues related to style

1. All of the references should be in brackets either ( ) or { } or [ ].

See above

Reviewer 2 Report

The authors, in this review article, tried to present the possible therapeutic approches for pediatirc patients to treat anogential warts and displayed an intresting case report. 

Abstract:
Line 23: 
No need for FDA abbreviation
Line 32: Use either HPV or Human papillomavirus, anogenital or genital warts and add other keywords

Introduction:
From line 35 to 85: The authors should diversify their references.

Line 35: Write Human papillomavirus in a single form;  Human papillomavirus belongs to

Line 63: Remove the word "to" before up to several months

Line 77:  The full name Human papillomavirus has been used in before in line 35.

Material and Methods:
Line 118: Use Age group better than population group.

Line 119: The full name of FDA has been mentioned before

Results:
The authors should add subtitles for each therapeutic modality under this section to allow the readers to easily follow up.

Line 138, 151: Check the order of reference 31

Line 159: Revise 6±12 months

Line 183: Replace therapy-unresponsive by resistant

I suggest that the authors should add some data related to immunotherapeutic apporoaches  including Candida, MMR, BCG,PPD,  HPV vaccines

Ca
se report:
Line 240:The full name of STI has been mentioned before.

Discussion:

Line 270:The full name of STI has been mentioned before.

Line 277: The information of "overall prevalence of AWs ranging from 0.2% to 5.1% at genital" has been mentioned before in line 269.

Line 293: Remove and before in the USA.

Line 323: Please explain the sentence "Girls are three times more interested than boys" 

This manuscript requires an English editing and grammer checks
